# Estimating the Effects of Sample Training Orders for Large Language Models without Retraining

## Abstract

The order of training samples plays a crucial role in large language models (LLMs), significantly impacting both their external performance and internal learning dynamics. Traditional methods for investigating this effect generally require retraining the model with various sample orders, which is computationally infeasible for LLMs. In this work, we improve traditional methods by designing a retraining-free framework. By approximating Adam optimizer updates with first- and second-order Taylor expansions and utilizing random projection methods to store intermediate checkpoints, our framework can efficiently estimate model parameters for arbitrary training sample orders. Next, we apply our framework to two downstream research problems: (1) Training curriculum design for LLMs — We base our retraining-free framework to propose a novel curriculum learning strategy that augments curriculum proposals with estimated model performances, enabling more informed sample scheduling. (2) LLMs' memorization and generalization effect analysis — We use our retraining-free framework to estimate how the positions of training samples influence LLMs' capacity for memorization and generalization. We conduct extensive experiments to validate the effectiveness of our retraining-free framework in reproducing the true model performances, and further demonstrate its potential in optimizing LLM training curricula and analyzing the memorization and generalization effects of LLMs.

## 1    Introduction

The order of training samples is crucial for optimizing large language models (LLMs), primarily due to the inherent nature of batch-based optimization methods (*e.g.*, mini-batch gradient descent) (Xue et al., 2023; Peng et al., 2025). This insight has spurred significant research in areas such as training curriculum design for LLMs, which strategically schedules training samples to enhance model optimization (Zhang et al., 2025b;a; Campos, 2021; Dai et al., 2025), and LLMs' memorization and generalization effect analysis (Lesci et al., 2024; Tirumala et al., 2022; Budnikov et al., 2025; Zheng & Jiang, 2022), which investigates how the sequence of sample exposure influences the model's ability to retain knowledge and generalize effectively. A straightforward strategy to study these problems is to train the target model multiple times with different sample orders, and then observe the results to either select the optimal one or analyze the underlying patterns (Zhang et al., 2018b; Xue et al., 2023; Kim & Lee, 2024).

In traditional machine learning, the above strategy is feasible because sample and parameter sizes are typically manageable, and training costs are relatively low (Zhang et al., 2018a; Graves et al., 2017). However, in the era of LLMs, this approach becomes impractical due to the massive scale of samples and parameters. This naturally raises a novel and fundamental research question:

*Can we estimate the effect of sample ordering on LLM performance without retraining?*

Despite its significance, answering this question is challenging. To begin with, a practical strategy for estimating model performance under a target sample order is to first measure the performance for a reference sample order and then infer the target performance by establishing a relationship between these two orders. However, since the target sample order can be arbitrary in an extremely

large space, identifying a common basis to effectively bridge the reference and target performances becomes a non-trivial challenge. And then, even if we can successfully identify a common basis for relating different sample orders, efficiently storing this basis also poses a significant challenge, as it may involve a vast number of LLM parameters.

To overcome the above challenges, in this paper, we propose a novel retraining-**f**ree framework by approximating the parameter **u**pdating process with **T**aylor expansions (called **FUT** for short). Specifically, we focus on the Adam optimizer and reformulate its update term as a function of the current model parameters. Next, we apply Taylor expansions to derive the relationships between the update terms across different model parameters based on the first- and second-order gradients of the loss function. This formulation establishes the common basis for correlating LLM performance across varying sample orders. Finally, we adopt the Random Projection based on the Johnson-Lindenstrauss (JL) theorem (Venkatasubramanian & Wang, 2011) to store the update terms for all training batches, significantly reducing memory consumption while maintaining accuracy.

Building on the above foundational framework, we further apply it to two specific research problems: (1) Training curriculum design for LLMs. Unlike traditional curriculum learning strategies that rely on human heuristics to determine sample orders, our framework empowers users to select sample orders based on the final model performance. Furthermore, for each sample order, our framework provides performance estimations, enabling users to make more informed decisions. (2) LLMs' memorization and generalization effect analysis. Unlike previous approaches that assess the impact of sample positioning on memorization and generalization through costly retraining or black-box neural network approximations, our framework offers an efficient and principled method to analyze these capabilities in LLMs.

In summary, the main contributions of this paper can be summarized as follows:

• We formally define the problem of "estimating the impact of training sample orders on model performance without retraining" in the context of LLMs.

• To solve the above problem, we propose a principled framework based on Taylor expansions and the Random Projection to efficiently estimate LLM performance for arbitrary sample orders.

• We apply our framework to two specific applications: (1) training curriculum design for LLMs and (2) LLMs' memorization and generalization effect analysis to demonstrate its fundamental nature and general applicability.

• We conduct extensive experiments to demonstrate the effectiveness of our framework in approximating the true performance and validate its potential in addressing the aforementioned applications.

## 2 PROBLEM FORMULATION

Suppose we have a training dataset with $T$ batches, denoted as $\mathcal{D}_{tr} = \{B_t\}_{t=0}^{T-1}$, and an LLM $M$. We begin by training $M$ on $\mathcal{D}_{tr}$ following a reference sample order and obtain the corresponding reference checkpoints[1]. Specifically, without loss of generality, we assume the reference sample order is $B_0, B_1, \ldots, B_{T-1}$, with the initial parameters of $M$ represented as $\theta_0$. After processing each batch $B_t$, the model parameters are updated from $\theta_t$ to $\theta_{t+1}$. Ultimately, we collect the reference checkpoints as $\Theta = \{\theta_t\}_{t=0}^T$. For a new sample order, $B_{l_0}, B_{l_1}, \ldots, B_{l_{T-1}}$, where $B_{l_t}$ is the $(t+1)$th training batch, our problem aims to efficiently derive the model parameters $\{\gamma_t\}_{t=0}^T$, where $\gamma_{t+1}$ is the model parameter after training batch $B_{l_t}$, and we set $\gamma_0 = \theta_0$.

**Relation with the influence function**. The above problem shares similarities with the influence function (Koh & Liang, 2017), as both study the effects of training samples. However, there are fundamental differences: our focus is on understanding the impact of sample ordering, while the influence function primarily examines the effect of removing individual samples. Moreover, our problem is situated within the context of LLMs, demanding efficient storage and management of large-scale model parameters.

**Straightforward solutions**. To solve the above problem, one is to retrain $M$ using the new sample order $B_{l_0}, B_{l_1}, \ldots, B_{l_{T-1}}$ and obtain the model parameters $\{\gamma_t\}_{t=0}^T$ after each batch. Another po-

---

[1]Note that the reference sample order can be arbitrary or chosen based on user preference.

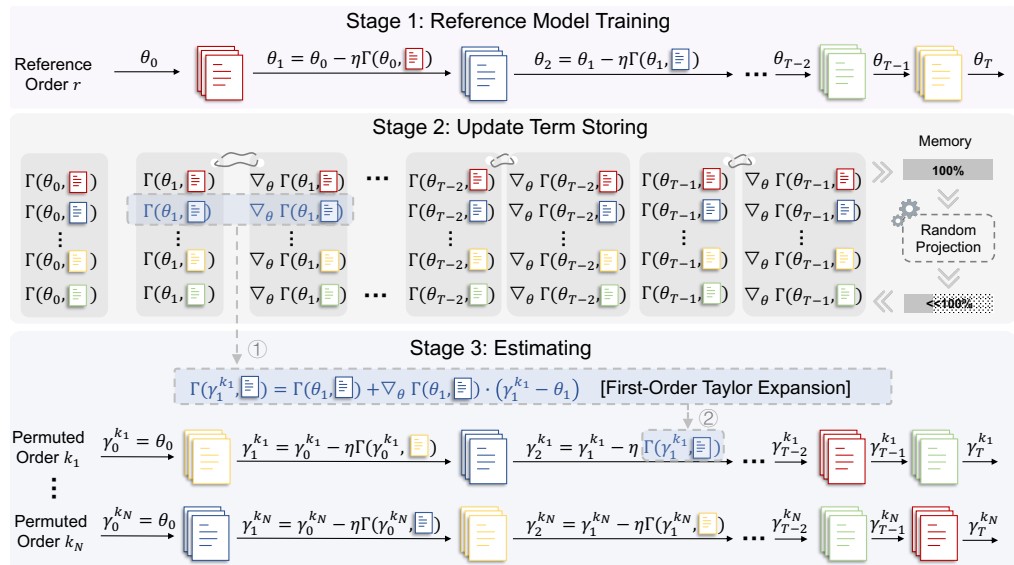

Figure 1: **Overview of the FUT framework.** FUT operates in three stages: **Stage 1:** Compute the reference trajectory $\Theta = \{\theta_t\}_{t=0}^{T}$ using a fixed data order $r$. **Stage 2:** Store update and gradient terms for all $(\theta_t, B_{l_t})$ pairs, compressing them via random projection. **Stage 3:** Estimate trajectories $\{\gamma_t^{k_i}\}_{t=0}^{T}$ under permuted data orders $\{k_i\}_{i=1}^{N}$ using first-order Taylor expansion based on stored terms. A toy example along the dashed line illustrates: ① retrieving stored terms for expansion, and ② updating parameters along a permuted order.

tential solution treats the sample order as the input to a neural network, with the model parameters as the output. In this way, a neural network could be trained to learn the correlation between the input and output, enabling parameter estimation without full retraining. However, the first solution demands substantial time and computational resources to retrain LLMs, rendering it practically infeasible. For the second solution, the limited availability of input-output pairs makes it difficult for a neural network to accurately learn the correlations, resulting in significantly lower performance.

## 3 THE FUT FRAMEWORK

To address the limitations of the above straightforward solutions, in this section, we propose a principled retraining-free framework. The core idea of our approach is to establish a relationship between $\{\gamma_t\}_{t=0}^{T}$ and $\{\theta_t\}_{t=0}^{T}$ by delving deeply into their respective generation processes. Then, we derive $\{\gamma_t\}_{t=0}^{T}$ based on $\{\theta_t\}_{t=0}^{T}$, which are precomputed as reference checkpoints.

Here, we focus on the Adam optimizer due to its widespread use in LLM optimization. However, our method can be easily extended to other batch-based gradient methods, such as SGD. By applying the updating rule of Adam, we have: [2]

$$\theta_{t+1} - \theta_t = -\eta\Gamma(\theta_t, B_t), \quad \forall\, 0 \leq t \leq T-1 \tag{1}$$

In this equation, $\Gamma(\theta_t, B_t) = m_t/(\sqrt{v_t} + \epsilon)$ is the update term, and

$$
\begin{aligned}
m_t &= (\beta_1 m_{t-1} + (1 - \beta_1)\nabla_\theta\mathcal{L}(\theta_t, B_t))/(1 - \beta_1^t), \\
v_t &= (\beta_2 v_{t-1} + (1 - \beta_2)\nabla_\theta\mathcal{L}(\theta_t, B_t)^2)/(1 - \beta_2^t),
\end{aligned}
\tag{2}
$$

where $\nabla_\theta\mathcal{L}(\theta_t, B_t)$ represents the gradient of the loss function $\mathcal{L}$ computed with respect to the model parameters $\theta_t$ using the mini-batch $B_t$. $\eta$ is the learning rate. $m_t$ and $v_t$ are the first and second momentum statistics, respectively. $\beta_1$ and $\beta_2$ are both the smoothing coefficients that control the decay rate of past gradients. $\epsilon$ is a small constant to prevent $m_t$ and $v_t$ from being divided by zero.

Similar to the above updating rule, we have $\gamma_{t+1} - \gamma_t = -\eta\Gamma(\gamma_t, B_{l_t})$ $(0 \leq t \leq T-1)$. To compute $\gamma_{t+1}$, we regard $\Gamma(\theta, B)$ as a function of the model parameters $\theta$. By using Taylor expansions on

---
[2]Without special mention, the updating is applied to each dimension of the parameter separately.

$\Gamma(\gamma_t, B_{l_t})$, we have:

$$\Gamma(\gamma_t, B_{l_t}) \approx \Gamma(\theta_t, B_{l_t}) + (\gamma_t - \theta_t)\nabla_\theta\Gamma(\theta_t, B_{l_t}) \tag{3}$$

where $\nabla_\theta\Gamma(\theta_t, B_{l_t})$ represents the gradient of $\Gamma(\theta_t, B_{l_t})$ with respect to $\theta$. In this equation, since $B_{l_t}$ is one of $B_0, B_1, \ldots, B_{T-1}$, if we can obtain $\Gamma(\theta_t, B_{l_t})$ and $\nabla_\theta\Gamma(\theta_t, B_{l_t})$ for all $0 \le t \le T-1$, then $\gamma_{t+1}$ can be recursively computed as follows:

$$\gamma_{t+1} = \gamma_t - \eta\Gamma(\theta_t, B_{l_t}) - \eta(\gamma_t - \theta_t)\nabla_\theta\Gamma(\theta_t, B_{l_t}), \tag{4}$$

where all the variables on the right-hand side are known. Here, $\Gamma(\theta_t, B_{l_t})$ and $\nabla_\theta\Gamma(\theta_t, B_{l_t})$ form the basis for connecting $\gamma_t$ and $\theta_t$. According to the Adam computational rules, we have:

$$\nabla_\theta\Gamma(\theta_t, B_{l_t}) = \frac{\frac{\partial m_t}{\partial \theta}(\sqrt{v_t} + \epsilon) - \frac{\partial \sqrt{v_t}}{\partial \theta}m_t}{(\sqrt{v_t} + \epsilon)^2} \tag{5}$$

where

$$\frac{\partial m_t}{\partial \theta} = \frac{\beta_1 \cdot \frac{\partial m_{t-1}}{\partial \theta} + (1 - \beta_1) \cdot \nabla_\theta^2\mathcal{L}(\theta_t, B_{l_t})}{1 - \beta_1^t},$$
$$\frac{\partial \sqrt{v_t}}{\partial \theta} = \frac{\beta_2 \cdot \frac{\partial v_{t-1}}{\partial \theta} + 2(1 - \beta_2) \cdot \nabla_\theta\mathcal{L}(\theta_t, B_{l_t}) \cdot \nabla_\theta^2\mathcal{L}(\theta_t, B_{l_t})}{2(1 - \beta_2^t)\sqrt{v_t}}. \tag{6}$$

By jointly observing equation (2) and (5), we can see $\Gamma(\theta_t, B_{l_t})$ and $\nabla_\theta\Gamma(\theta_t, B_{l_t})$ only rely on $\nabla_\theta\mathcal{L}(\theta_t, B_{l_t})$ and $\nabla_\theta^2\mathcal{L}(\theta_t, B_{l_t})$. These terms are the gradients of the loss function with respect to the reference checkpoint and the training batch. Since the reference checkpoints $\{\theta_t\}_{t=0}^T$ have already been collected before, we can efficiently compute $\nabla_\theta\mathcal{L}(\theta_t, B_{l_t})$ and $\nabla_\theta^2\mathcal{L}(\theta_t, B_{l_t})$ simply by bringing $\theta_t$ and $B_{l_t}$ into the gradient functions.

The algorithm for deriving $\{\gamma_t\}_{t=0}^T$ is shown in Algorithm 1. In specific, there are three stages. In the reference model training stage, we train $M$ using $\mathcal{D}_{tr}$ based on the reference sample order. After obtaining $\Theta = \{\theta_t\}_{t=0}^T$, in the update term storing stage, we derive and store $\Gamma(\theta_t, B_{l_t})$ and $\nabla_\theta\Gamma(\theta_t, B_{l_t})$ for all $0 \le t \le T-1$ based on equation (2) and (5). At last, in the estimation stage, for a new sample order $\{l_t\}_{t=0}^{T-1}$, we compute $\{\gamma_t\}_{t=0}^T$ based on equation (4) in a recursive manner. In practice, the first two stages are executed only once, after which the performance of any new sample order can be efficiently estimated. Figure 1 illustrates the complete FUT framework.

**Enhanced model with the second-order Taylor expansion**. In the above method, we approximate $\Gamma(\gamma_t, B_{l_t})$ with the first-order Taylor expansion. To enhance accuracy, we extend our approach by incorporating the second-order term, resulting in an updated version of equation (3) as follows:

$$\Gamma(\gamma_t, B_{l_t}) \approx \Gamma(\theta_t, B_{l_t}) + (\gamma_t - \theta_t)\nabla_\theta\Gamma(\theta_t, B_{l_t}) + \frac{1}{2} \cdot (\gamma_t - \theta_t)^2\nabla_\theta^2\Gamma(\theta_t, B_{l_t}) \tag{7}$$

where $\nabla_\theta^2\Gamma(\theta_t, B_{l_t})$ is the second-order gradient of $\Gamma(\theta_t, B_{l_t})$. By combining this equation with $\gamma_{t+1} - \gamma_t = -\eta\Gamma(\gamma_t, B_{l_t})$, we have:

$$\gamma_{t+1} = \gamma_t - \eta\Gamma(\theta_t, B_{l_t}) - \eta(\gamma_t - \theta_t)\nabla_\theta\Gamma(\theta_t, B_{l_t}) - \frac{1}{2}\eta \cdot (\gamma_t - \theta_t)^2\nabla_\theta^2\Gamma(\theta_t, B_{l_t}). \tag{8}$$

Please referred to Appendix B.1 for more details to precompute $\nabla_\theta^2\Gamma(\theta_t, B_{l_t})$. After obtaining $\nabla_\theta^2\Gamma(\theta_t, B_{l_t})$, we can efficiently derive $\{\gamma_t\}_{t=0}^T$ based on equation (8) in a recursive manner.

**Efficient storage of the update terms**. According to the above analysis, our framework heavily rely on $\Gamma(\theta_t, B_{l_t})$, $\nabla_\theta\Gamma(\theta_t, B_{l_t})$ and $\nabla_\theta^2\Gamma(\theta_t, B_{l_t})$. However, in the context of LLMs, their dimensions are extremely large, posing significant storage challenges. To address this issue, we leverage the Random Projection technique (Chen et al., 2019a; Zhang et al., 2018b) based on the Johnson-Lindenstrauss (JL) theorem (Venkatasubramanian & Wang, 2011) to efficiently reduce their dimensionality. To illustrate this process, consider storing a 2-dimensional matrix $M \in R^{d_1 \times d_2}$. We first generate a random matrix $A \in R^{d_2 \times k}$ that follows a Gaussian distribution $\mathcal{N}(0, 1/k)$, where $k$ is the target dimension chosen based on the JL theorem. Next, we perform dimensionality reduction by left-multiplying $A$ with $M$, that is, $M' = MA$. Here, $M' \in R^{d_1 \times k}$ is the compressed representation for storage. To recover the original matrix $M$, we similarly perform a left multiplication using the Moore-Penrose pseudoinverse of $A$, denoted as $A^+$, that is, $\widetilde{M} = M'A^+$. This

---

**Algorithm 1** FUT Framework for Deriving $\{\gamma_t\}_{t=0}^T$ with First-order Taylor Expansion

---

**Require:** Initialized model parameter $\theta_0$, reference training batches $\{B_t\}_{t=0}^{T-1}$, learning rate $\eta$, and $\epsilon$.
**Ensure:** Derived sequence $\{\gamma_t\}_{t=0}^T$
  1: Reference Model Training Stage:
  2: **for** $t = 0$ to $T - 1$ **do**
  3:     Compute the $(t + 1)$th reference checkpoint:
  4:         $\theta_{t+1} \leftarrow \theta_t - \eta\Gamma(\theta_t, B_t)$    (Eq. 1)
  5: **end for**
  6: Obtain $\Theta = \{\theta_t\}_{t=0}^T$
  7: Update Term Storing Stage:
  8: **for** $t = 0$ to $T - 1$ **do**
  9:     Compute first- and second-order update terms:
 10:         $\Gamma(\theta_t, B_{l_t}) \leftarrow$ calculate $\nabla_\theta\mathcal{L}(\theta_t, B_{l_t})$ with checkpoint $\theta_t$ on batch $B_{l_t}$
 11:         $\nabla_\theta\Gamma(\theta_t, B_{l_t}) \leftarrow$ calculate $\nabla_\theta\mathcal{L}(\theta_t, B_{l_t}), \nabla_\theta^2\mathcal{L}(\theta_t, B_{l_t})$ with checkpoint $\theta_t$ on batch $B_{l_t}$
 12: **end for**
 13: Estimation Stage:
 14: $\gamma_0 \leftarrow \theta_0$
 15: **for** $t = 0$ to $T - 1$ **do**
 16:     First-order Taylor expansion for $\Gamma(\gamma_t, B_{l_t})$:
 17:         $\Gamma(\gamma_t, B_{l_t}) \leftarrow \Gamma(\theta_t, B_{l_t}) + (\gamma_t - \theta_t)\nabla_\theta\Gamma(\theta_t, B_{l_t})$    (Eq. 3)
 18:     Update $\gamma_{t+1}$:
 19:         $\gamma_{t+1} \leftarrow \gamma_t - \eta\Gamma(\gamma_t, B_{l_t})$    (Eq. 4)
 20: **end for**
 21: **Return** $\{\gamma_t\}_{t=0}^T$

---

approach effectively reduces the space complexity of $M$ from $\mathcal{O}(d_1 d_2)$ to $\mathcal{O}(d_1 k)$, where $k \ll d$, significantly alleviating the storage burden when precomputing it. For higher-order terms such as $\nabla_\theta\Gamma(\theta_t, B_{l_t})$ and $\nabla_\theta^2\Gamma(\theta_t, B_{l_t})$, we similarly apply the random projection technique to reduce their storage complexity, making the process efficient and scalable.

**Comparison between the computational costs of retraining and our method**. Assume that the time complexity for computing the loss gradient once is $\mathcal{O}(C)$. Enumerating model parameters under all possible training orders requires retraining the model on the original dataset for $T!$ times, where in each permuted order, we need to perform $\nabla_\theta\mathcal{L}(\theta_t, B_{l_t})$ for $T$ times. Therefore, the total time complexity of retraining is $\mathcal{O}(T \cdot C \cdot T!)$, which is computationally prohibitive for LLMs with billions of parameters. In contrast, our method estimates the model updates under different batch orders without retraining. Its main computational cost comes from computing the updating terms $\Gamma(\theta_t, B_{l_t})$, $\nabla_\theta\Gamma(\theta_t, B_{l_t})$, and $\nabla_\theta^2\Gamma(\theta_t, B_{l_t})$. In specific, each of these terms requires a single backward computation of the model at checkpoint $\theta_t$ over batch $B_{l_t}$, *i.e.*, $\mathcal{L}(\theta_t, B_{l_t})$. Since there are $T^2$ such $(\theta_t, B_{l_t})$ pairs in total, the overall time complexity of our method is $\mathcal{O}(T^2 \cdot C)$.

## 4 APPLICATIONS

### 4.1 TRAINING CURRICULUM DESIGN FOR LLMS

**Problem definition**. Following the notations in Section 2, suppose we aim to train a model $M$ on the dataset $\mathcal{D}_{tr} = \{B_t\}_{t=0}^{T-1}$. Let $\pi$ be a permutation function that maps the standard index set $\{0, 1, \dots, T - 1\}$ to $\{\pi(0), \pi(1), \dots, \pi(T - 1)\}$, where $\pi(t) \in [0, T - 1]$ indicates that batch $B_t$ is placed at the $(\pi(t) + 1)$-th position in the training sequence. Following common practice, we train the LLM for only one epoch (Xue et al., 2023). The goal is to find an optimal permutation $\pi^*$ such that the resulting model performs best on a validation set $\mathcal{D}_{val}$, formally defined as:

$$\pi^* := \underset{\pi \in \Pi}{\arg\max}\, \mathcal{R}(\gamma_T^\pi, \mathcal{D}_{val}), \tag{9}$$

where $\gamma_T^\pi$ denotes the final model parameters estimated using our FUT framework, and the training order $l_t$ is induced by $\pi$. The performance metric $\mathcal{R}$ is implemented using Perplexity (PPL)Hu et al. (2024), and $\Pi$ denotes the space of all possible permutation functions.

**Our solution based on FUT**. Since objective (9) is non-differentiable, we design a Genetic Algorithm (GA) (Katoch et al., 2020) to obtain $\pi^*$. In specific, we maintain a set of candidate sample orders and iteratively apply crossover and mutation operators to generate improved sample orders, aiming to optimize the model performance. For more details, we refer readers to Appendix B.3.

Compared to traditional curriculum learning strategies, a key advantage of our method is its ability to estimate model performance for each curriculum proposal, enabling more informed decisions. For instance, by knowing the performance gap between different curricula, users can assess whether the difference is significant. If the gap is small, users can confidently choose one at random.

## 4.2 LLMs' Memorization and Generalization Effect Analysis

**Problem definition**. We continue to follow the notations introduced in Section 2. For each training batch in $\mathcal{D}_{tr}$, the memorization problem evaluates model performance when the batch appears at different positions in the training sequence. Specifically, we use the following evaluation method:

$$M_{i,j} = \frac{1}{N} \sum_{k=1}^{N} \mathcal{R}(\theta_T^{\pi_k^{ij}}, B_i),$$

where $\pi_k^{ij}$ is a permutation function that fixes $B_i$ at the $j$-th training position while randomly shuffling all other batches. For each $B_i$, we generate $N$ such permutations, and the final performance is computed as the average across these permutations. The generalization problem is defined in a similar manner, with the key distinction that $B_i$ in the above equation is replaced by $D_i$, a dataset not seen during training, i.e., $D_i \notin \mathcal{D}_{tr}$.

**Our solution based on FUT**. For each $\pi_k^{ij}$, we first generate the sequence $l_t$ and then estimate $\gamma_T^{\pi_k^{ij}}$ using the reference checkpoints $\{\theta_t\}_{t=0}^T$. Finally, we compute $\mathcal{R}(\theta_T^{\pi_k^{ij}}, B_i)$ or $\mathcal{R}(\theta_T^{\pi_k^{ij}}, D_i)$ based on $\gamma_T^{\pi_k^{ij}}$, and average the resulting performances over different values of $k$.

Compared to previous studies that estimate memorization capability using black-box neural network (Zheng & Jiang, 2022; Lesci et al., 2024; Feldman & Zhang, 2020), our method is more principled and grounded in theoretical foundations.

## 5 Experiments

In this section, we conduct extensive experiments to demonstrate the effectiveness of our framework and its potential applications in designing LLM training curricula and analyzing LLMs' memorization and generalization capabilities.

### 5.1 Evaluation on the General Capability of Our Methods

**Experimental Setup**. To evaluate the effectiveness of our FUT framework, we incorporate the estimated model parameters into the LLM and measure the performance gap between the estimated and actual results. Specifically, we conduct our experiments on the Wikitext dataset (Merity et al., 2018a), a curated collection of high-quality English Wikipedia articles that is widely used for language modeling and evaluation. This dataset is particularly well-suited for assessing model perplexity due to its long-range token dependencies (Merity et al., 2018b). We adopt the architecture of LLaMA (Touvron et al., 2023) to construct a base model with 636 million parameters. The model has a hidden size of 2048 and consists of 10 stacked transformer layers with 10 attention heads. We choose this relatively small architecture because our main experiments involve repeated LLM training to validate that the proposed FUT framework can accurately estimate model parameters under various training orders. In Appendix D.1, we scale the model size up to 6.0 billion parameters to assess the scalability of our approach. Following common practice (Xue et al., 2023), we use the Adam optimizer for LLM training and train for a single epoch to evaluate performance based on perplexity (Hu et al., 2024). We also report fine-grained performance estimation results at intermediate stages in Appendix D.3.

In our experiments, assuming the dataset consists of $T$ batches, we randomly select $N$ training orders from the total of $T!$ possible permutations. For each selected order, we use our method to

Table 1: Performance estimation accuracy (use AbsDiff as metric) across methods under settings with different numbers of batches.

| T | Random | FUT | FUT++ |
|---|--------|-----|-------|
| 8 | 0.0205 | 0.0165 | **0.0085** |
| 16 | 0.0917 | **0.0649** | 0.0703 |
| 32 | 0.0373 | 0.0290 | **0.0193** |
| 64 | 0.0644 | 0.0445 | **0.0319** |
| 128 | 0.0575 | 0.0372 | **0.0284** |
| 256 | 0.0471 | **0.0205** | 0.0368 |

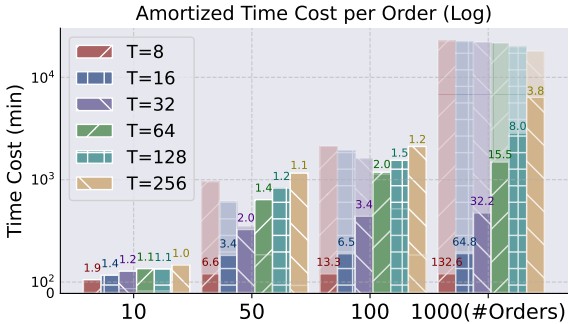

Figure 2: Time cost comparison. The numbers indicate how many times our model time exceeds Retraining.

estimate the model performance $\hat{r}$ and also train the LLM using that order to obtain the ground-truth performance $r$. The performance gap is then calculated as: **AbsDiff** $= \frac{1}{N} \sum_{k=1}^{N} |\hat{r}_k - r_k|$, where $k$ indexes the different training orders. We set $N = 10$ to balance evaluation reliability with computational cost. To assess the scalability and robustness of our framework, we vary the number of batches $T$ across the set $\{8, 16, 32, 64, 128, 256\}$. This setup allows us to evaluate how performance estimation behaves under increasing training granularity and longer optimization trajectories.

**Baseline**. We denote our method using the first- and second-order Taylor expansions as **FUT** and **FUT++**, respectively. The ground-truth performance obtained via actual LLM training is referred to as **Retraining**. Additionally, we introduce a heuristic baseline, named **Random**, where we first obtain all the $N$ ground-truth performances $\{r_k\}_{k=1}^{N}$, and then randomly estimate the performance within the range $\left[\min_{k \in [1,N]} r_k, \ \max_{k \in [1,N]} r_k\right]$.

**Results**. Table 1 shows that the Random baseline performs the worst, indicating that estimating LLM performance without retraining itself is a non-trivial task. Both FUT and FUT++ consistently outperform Random across all batch settings with considerable margins, demonstrating their effectiveness. Between our methods, FUT++ performs better than FUT in more cases, suggesting that the inclusion of the second-order term in the Taylor expansion is beneficial for our problem. In addition, we also compare the efficiency of our method with the Retraining strategy. Here, we vary $N$ over the set $\{10, 50, 100, 1000\}$ to observe the trend as the number of sample orders increases[3]. We compare different methods with various $T$'s. The results are presented in Figure 2, where the solid bars represent our method and the dashed bars represent Retraining. We observe that as the total number of orders increases, our method progressively achieves higher time efficiency per order compared to Retraining, with a maximum speedup of 132.6 times. Our methods across all $T$ surpass Retraining, highlighting the advantages of our methods in scalability. We also conduct experiments to demonstrate the necessity of using the random projection for storage in Appendix D.2.

## 5.2 EVALUATION ON THE APPLICATION OF TRAINING CURRICULUM DESIGN FOR LLMS

**Experimental Setup & Baselines**. In this experiment, we evaluate whether our methods can assist in designing more effective training curricula for LLMs. Similar to the above section, we use perplexity as the evaluation metric and measure different models by varying $T$ in the range of $\{8, 16, 32, 64, 128, 256\}$. We compare our methods with the following baselines:

• **Random Order (RO)**, which generates the curriculum by randomly shuffling the training batches.

• **Sample Length (SL)** (Campos, 2021), which is a difficulty-based curriculum design strategy, and the difficulty score is determined based on the sentence length.

• **Perplexity (PPL)** (Zhang et al., 2025a), which uses the perplexity from a reference model as a proxy to evaluate sample difficulty and design the curriculum.

• **Perplexity Difference (PD)** (Zhang et al., 2025b), measuring the perplexity gap between a strong and a weak model, treating samples with larger gaps as more difficult to design the curriculum.

---

[3]The costs of the first two stages in our method are amortized across all sample orders, as they are executed only once.

Table 2: Perplexity results across different batch numbers and curriculum design strategies. "Est." means the performance estimated by our methods, and the colored results represent the best estimation accuracy between the FUT and FUT++ methods.

| Methods | RO | Len | PPL | PD | FUT (Est.) | | FUT++ (Est.) | |
|---|---|---|---|---|---|---|---|---|
| 8 | 1.4414 | 1.4392 | 1.4012 | 1.4006 | **1.3996** | (1.3963) | 1.3998 | (1.3962) |
| 16 | 1.4599 | 1.5291 | 1.4531 | 1.4542 | 1.4536 | (1.4314) | **1.4523** | (1.4307) |
| 32 | 1.4109 | 1.4042 | 1.3966 | 1.3933 | 1.3909 | (1.3823) | **1.3881** | (1.3686) |
| 64 | 1.4248 | 1.4079 | 1.4027 | 1.4071 | **1.3785** | (1.3838) | 1.3804 | (1.3856) |
| 128 | 1.3838 | 1.3872 | 1.3790 | 1.3697 | **1.3412** | (1.3446) | 1.3619 | (1.3512) |
| 256 | 1.3696 | 1.3766 | 1.3645 | 1.3660 | 1.3378 | (1.3551) | **1.3178** | (1.3460) |

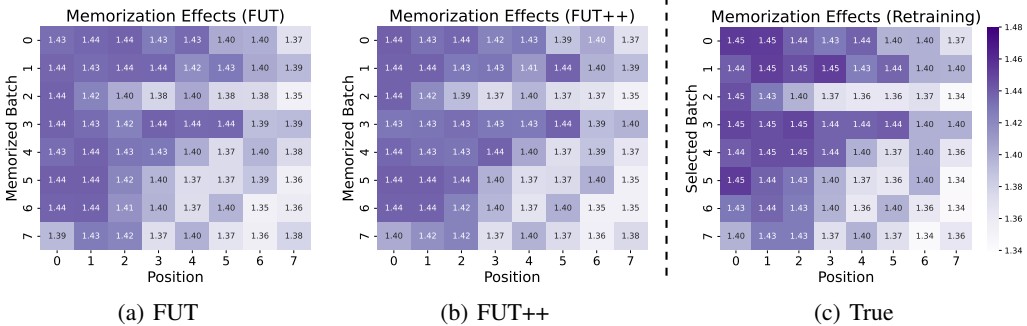

(a) FUT  (b) FUT++  (c) True

Figure 3: Memorization effects. Heatmaps in (a) and (b) are estimated by our FUT and FUT++ methods, respectively. Heatmap in (c) represents the true memorization effect obtained by retraining.

We use the baseline methods and our proposed approaches (using equation (9)) to generate training curricula, and train the LLM based on them for comparison.

**Results**. The results are shown in Table 2. We can see: In most cases, **RO** performs the worst, as it lacks any problem-specific design and simply generates the training curriculum randomly. **PPL** and **PD** consistently outperform **SL** across different batch sizes, which is as expected since they both leverage perplexity as a proxy to design the curricula-aligning well with the final evaluation metric. Finally, our methods achieve superior performance compared to all baselines, demonstrating their effectiveness in designing training curricula for LLMs. As shown in the last two columns of Table 2, our methods provide performance estimates that closely match the actual results, enabling more efficient decision-making when selecting optimal training orders.

### 5.3 EVALUATION ON THE APPLICATION OF LLM MEMORIZATION & GENERALIZATION EFFECT ANALYSIS

**Experimental Setup**. In this experiment, we evaluate the memorization & generalization effects of LLM when a sample batch is placed at different training positions. In specific, the number of training batches is set as 8 (*i.e.*, $T = 8$). We visualize the value of $M_{i,j}$ in Section 4.2 based on perplexity by setting different $(i, j)$ pairs.

**Results**. The results are presented in Figure 3, 4 and 5. We can see: Compared to the true memorization effect (in Figure 3(c)), where we retrain the LLM to compute $M_{i,j}$, FUT and FUT++ in Figure 3(a) and (b), can accurately estimate the model's memorization of different batches at various positions using both first- and second-order approximations, respectively. The results reveal that the model tends to memorize batches appearing later in the training order more effectively, as indicated by lower perplexity. For generalization analysis, we divide training batches into two groups based on their similarity to the test set $D$, using the average similarity $\tau$ as a threshold. Our metric is the cosine similarity of the sample representations. As shown in Figures 4 and 5, our method (dashed red/blue lines) closely estimates the true performance (black line) and captures the same generalization trend in most cases. In Figure 4, batches similar to the test data generalize better when placed

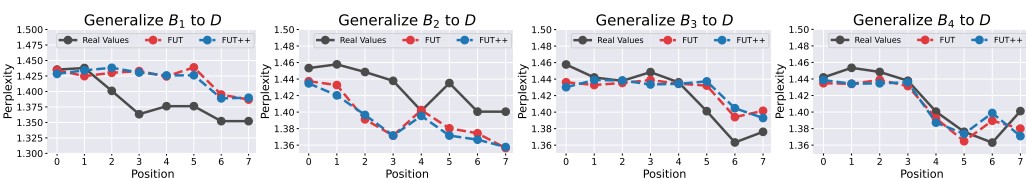

Figure 4: The generalization effect of batch $B_i$ on dataset $D$, with $\text{sim}(B_i, D) >= \tau$.

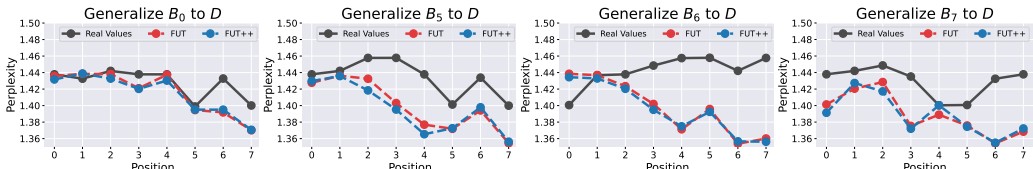

Figure 5: The generalization effect of batch $B_i$ on dataset $D$, with $\text{sim}(B_i, D) < \tau$.

later in training. In contrast, Figure 5 shows that dissimilar batches have little or random effect on generalization, regardless of their positions in the training sequences.

## 6 RELATED WORK

**Training Dynamics of Language Models.** Understanding training dynamics is essential for analyzing how deep models evolve during optimization (Frankle et al., 2020; Raghu et al., 2017; Achille et al., 2018). In the context of language models, early work focused on the evolution of learned representations (Saphra & Lopez, 2018; Saphra, 2021) and the encoding of world knowledge (Liu et al., 2021) during pre-training. These insights have also been extended to downstream tasks such as summarization (Goyal et al., 2022) and speech translation (Savoldi et al., 2022). More recent studies have begun to examine the training dynamics of LLMs (Ren & Sutherland, 2025; Biderman et al., 2023; Teehan et al., 2022; Lesci et al., 2024; Dai et al., 2025), which are harder to analyze due to their scale. For example, Teehan et al. (2022) studies internal representation development and structural changes during training, while Biderman et al. (2023) uses models of varying sizes to study how training behavior shifts with scale. Additionally, Ren & Sutherland (2025) explores how learning certain examples affects the model's behavior on other inputs.

**Influence Function**. Influence function is used to estimate the impact of each training sample on a specific test prediction (Koh & Liang, 2017; Bae et al., 2022; Koh et al., 2019). The foundational work by Koh & Liang (2017) applies influence functions by calculating gradients and Hessian-vector products to measure the contribution of each training example to a test point. However, research in (Basu et al., 2021; Guo et al., 2021) has shown that influence functions can be unstable and unreliable in neural network. Additionally, computing the necessary Hessian-vector products is computationally expensive, particularly for LLMs. To address this challenge, a recent study by (Lin et al., 2024) introduces a caching mechanism to estimate token-level influences in LLMs. While this method alleviates some computational difficulties, it overlooks the crucial influence of sample order in the training process, which plays a significant role in shaping the learning dynamics.

## 7 CONCLUSION

In this work, we propose a retraining-free framework for analyzing the effect of training sample order on LLMs, addressing the prohibitive cost of retraining-based approaches. By approximating the optimization dynamics of Adam via Taylor expansion and employing random projection for efficient parameter estimation, our framework enables accurate performance prediction under arbitrary sample orders. We demonstrate the utility of this framework in two key research problems of LLMs: training curriculum design, and memorization & generalization effect analysis. Extensive experiments show that our framework faithfully approximates true model performance and provides valuable insights into both external performance and internal learning dynamics of LLMs. Our framework offers a practical tool for understanding and optimizing the model behaviors of LLMs.

ETHICS STATEMENT

We confirm that this research adheres to the ICLR Code of Ethics. Our study does not involve human subjects, and all datasets used are publicly available, with no sensitive or personal data involved. We have considered the ethical implications of our work, particularly in terms of fairness and potential misuse of large language models. We are committed to ensuring that our research contributes positively to the academic community and society at large.

REPRODUCIBILITY STATEMENT

To ensure reproducibility of our results, we provide clear descriptions of the methods and experiments in the main text and appendix. The framework used in our experiments is built upon the Adam optimizer with first- and second-order Taylor expansions, which are explained in detail in Section 3, particularly in Algorithm 1. The source code for the implementation of our retraining-free framework, as well as additional resources, are available as supplementary materials. For datasets, we use the Wikitext dataset as detailed in Section 5.1. All experimental settings and hyperparameters used in our evaluation are provided in Appendix C to ensure that our results can be reproduced accurately.

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

CONTENTS

## A    ACKNOWLEDGMENT OF LLM USAGE

For this manuscript, large language models (LLMs) were only used to assist with language editing, including the correction of typos, enhancement of grammar, and improvement of phrasing. They were not involved in any aspect of research ideation, analysis, result generation, or interpretation. The responsibility for all scientific content in this work rests entirely with the authors.

## B    TECHNICAL DETAILS

### B.1    PRECOMPUTATION IN UPDATE TERM STORING STAGE

Recall that the proposed FUT framework consists of three stages in Figure 1, where the update term storing (Stage 2) plays an important role to bridge the gap between the learning dynamic of reference order and that of the new order. In specific, in Stage 2, we need to compute three kinds of update terms: $\Gamma(\theta_t, B_{l_t})$ and $\nabla_\theta \Gamma(\theta_t, B_{l_t})$ for the first-order Taylor expansion (in equation (3)), and the additional $\nabla_\theta^2 \Gamma(\theta_t, B_{l_t})$ for the second-order Taylor expansion (in equation (7)). In the following, we describe how we compute these three update terms in detail, respectively.

• First, for each $\Gamma(\theta_t, B_{l_t})$ term, we can compute it by directly applying the checkpoint $\theta_t$ over batch $B_{l_t}$ following the updating rule of Adam optimizer:

$$\Gamma(\theta_t, B_{l_t}) = m_t / (\sqrt{v_t} + \epsilon), \tag{10}$$

where

$$m_t = (\beta_1 \cdot m_{t-1} + (1 - \beta_1) \cdot \nabla_\theta \mathcal{L}(B_{l_t}; \theta_t)) / (1 - \beta_1^t),$$
$$v_t = (\beta_2 \cdot v_{t-1} + (1 - \beta_2) \cdot \nabla_\theta \mathcal{L}(B_{l_t}; \theta_t)^2) / (1 - \beta_2^t), \tag{11}$$

where the accumulative terms $m_{t-1}$ and $v_{t-1}$ terms in $m_t$ and $v_t$ are constructed by the gradient from the last step in the original training process, *i.e.*, $\nabla_\theta \mathcal{L}(B_{t-1}; \theta_{t-1})$.

• Second, for each first-order update term $\nabla_\theta \Gamma(\theta_t, B_{l_t})$, we first expand it as:

$$\nabla \Gamma(\theta_t, B_{l_t}) = \frac{\partial \Gamma(\theta_t, B_{l_t})}{\partial \theta} = \frac{\frac{\partial m_t}{\partial \theta} (\sqrt{v_t} + \epsilon) - \frac{\partial \sqrt{v_t}}{\partial \theta} m_t}{(\sqrt{v_t} + \epsilon)^2} \tag{12}$$

where

$$\frac{\partial m_t}{\partial \theta} = \frac{\beta_1 \cdot \frac{\partial m_{t-1}}{\partial \theta} + (1 - \beta_1) \cdot \nabla_\theta^2 \mathcal{L}(B_{l_t}; \theta_t)}{1 - \beta_1^t},$$
$$\frac{\partial \sqrt{v_t}}{\partial \theta} = \frac{\beta_2 \cdot \frac{\partial v_{t-1}}{\partial \theta} + 2(1 - \beta_2) \cdot \nabla_\theta \mathcal{L}(B_{l_t}; \theta_t) \cdot \nabla_\theta^2 \mathcal{L}(B_{l_t}; \theta_t)}{2(1 - \beta_2^t)\sqrt{v_t}}, \tag{13}$$

To compute the second-order gradient $\nabla_\theta^2 \mathcal{L}(B_{l_t}; \theta_t)$, a straightforward approach is to apply the backward operator to $\mathcal{L}(B_{l_t}; \theta_t)$ twice. However, this requires computing the Hessian matrix of the parameters, which is prohibitively expensive, especially for LLMs with a large number of parameters.

To address this limitation, we approximate the second-order gradient using $(\nabla_\theta \mathcal{L}(B_{l_t}; \theta_t) - \nabla_\theta \mathcal{L}(B_{l_t}; \theta_{t-1})) / (\theta_t - \theta_{t-1})$, where $\theta_{t-1}$ denotes the parameter at step $t-1$ in the original training process. This approximation is justified by the limited variation in parameter updates between adjacent training steps.

• At last, for each $\nabla^2 \Gamma(\theta_t, B_{l_t})$ term, we can also expand it as:

$$\nabla^2 \Gamma(\theta_t, B_{l_t}) = \frac{\partial^2 \Gamma(\theta_t, B_{l_t})}{\partial \theta^2} = \frac{1}{(\sqrt{v_t} + \epsilon)^2} \left[ \left( \frac{\partial^2 m_t}{\partial \theta^2} \right) (\sqrt{v_t} + \epsilon) - \left( \frac{\partial^2 \sqrt{v_t}}{\partial \theta^2} \right) m_t \right.$$
$$\left. - 2 \left( \frac{\partial \sqrt{v_t}}{\partial \theta} \right) \left( \frac{\partial m_t}{\partial \theta} \right) + 2 \left( \frac{\partial \sqrt{v_t}}{\partial \theta} \right)^2 \frac{m_t}{\sqrt{v_t} + \epsilon} \right] \tag{14}$$

where

$$\frac{\partial^2 m_t}{\partial \theta^2} = \frac{\beta_1 \cdot \frac{\partial^2 m_{t-1}}{\partial \theta^2} + (1-\beta_1) \cdot \nabla_\theta^3 \mathcal{L}(B_{l_t}; \theta_t)}{1 - \beta_1^t},$$

$$\frac{\partial^2 \sqrt{v_t}}{\partial \theta^2} = \frac{\beta_2 \cdot \frac{\partial^2 v_{t-1}}{\partial \theta^2} + 2(1-\beta_2)\left[\nabla_\theta^2 \mathcal{L}(B_{l_t}; \theta_t) \cdot \nabla_\theta^2 \mathcal{L}(B_{l_t}; \theta_t) + \nabla_\theta \mathcal{L}(B_{l_t}; \theta_t) \cdot \nabla_\theta^3 \mathcal{L}(B_{l_t}; \theta_t)\right]}{2(1 - \beta_2^t)\sqrt{v_t}}$$

$$- \frac{\left(\beta_2 \cdot \frac{\partial v_{t-1}}{\partial \theta} + 2(1-\beta_2) \cdot \nabla_\theta \mathcal{L}(B_{l_t}; \theta_t) \cdot \nabla_\theta^2 \mathcal{L}(B_{l_t}; \theta_t)\right) \cdot \frac{\partial v_t}{\partial \theta}}{4(1 - \beta_2^t)(v_t)^{3/2}},$$

$$(15)$$

Similarly, to compute the third-order gradient $\nabla_\theta^3 \mathcal{L}(B_{l_t}; \theta_t)$, we use $(\nabla_\theta^2 \mathcal{L}(B_{l_t}; \theta_t) - \nabla_\theta^2 \mathcal{L}(B_{l_t}; \theta_{t-1}))/(\theta_t - \theta_{t-1})$ to approximate it.

By computing these update terms for each $(\theta_t, B_{l_t})$ pair, we can access all the update terms we may need in Estimating Stage (Stage 3 in Figure 1). That is, given an arbitrary permuted order, which is different from the reference one, we can recursively execute the first-order Taylor expansion in equation (3) or the second-order Taylor expansion in equation (7) to obtain the new model parameters.

## B.2 RANDOM PROJECTION FOR STORING UPDATE TERMS

The update terms $\Gamma(\theta_t, B_{l_t})$, $\nabla_\theta \Gamma(\theta_t, B_{l_t})$, and $\nabla_\theta^2 \Gamma(\theta_t, B_{l_t})$ are essential to our FUT framework. However, in large-scale neural networks such as LLMs, these terms typically have dimensionality comparable to that of the model parameters, making direct precomputation and storage for every pair $(\theta_t, B_{l_t})$ prohibitively expensive in terms of memory.

To mitigate this issue, we adopt a random projection strategy based on the Johnson–Lindenstrauss (JL) theorem (Venkatasubramanian & Wang, 2011), following the well-established compression techniques in (Lin et al., 2024). The JL theorem guarantees that high-dimensional vectors can be embedded into a significantly lower-dimensional space with bounded distortion of pairwise distances, which aligns well with our goal of efficiently storing approximate versions of gradient-related terms.

**Theorem 1 (Johnson–Lindenstrauss Theorem)** *Let $0 < \epsilon < 1$ and let $X = \{x_1, x_2, \ldots, x_n\} \subset \mathbb{R}^d$ be a set of $n$ vectors. Then there exists a linear mapping $f : \mathbb{R}^d \to \mathbb{R}^k$, where $k = \mathcal{O}(\epsilon^{-2} \log n)$, such that for all $x_i, x_j \in X$,*

$$(1-\epsilon)\|x_i - x_j\|_2^2 \le \|f(x_i) - f(x_j)\|_2^2 \le (1+\epsilon)\|x_i - x_j\|_2^2.$$

In our setting, we apply the JL projection to compress each update matrix prior to storage. Formally, for any matrix $M \in \mathbb{R}^{d_1 \times d_2}$—where $M$ may represent $\Gamma(\theta_t, B_{l_t})$, $\nabla_\theta \Gamma(\theta_t, B_{l_t})$, or $\nabla_\theta^2 \Gamma(\theta_t, B_{l_t})$—we generate a random projection matrix $A \in \mathbb{R}^{d_2 \times k}$ whose entries are sampled i.i.d. from a Gaussian distribution: $A_{ij} \sim \mathcal{N}(0, 1/k)$. The compressed representation of $M$ is then given by:

$$M' = MA \in \mathbb{R}^{d_1 \times k}.$$

This projection reduces the space complexity from $\mathcal{O}(d_1 d_2)$ to $\mathcal{O}(d_1 k)$ while approximately preserving the geometric structure of the original matrix rows.

To recover these terms for estimating the parameters under a new batch order, an approximate reconstruction can be achieved using the Moore–Penrose pseudoinverse $A^+ \in \mathbb{R}^{k \times d_2}$ as:

$$\widetilde{M} = M'A^+ \approx M.$$

In practice, the target dimension $k$ is selected based on the number of rows $d_1$ in $M$, which corresponds to the number of vectors $n$ in Theorem 1. To balance accuracy and memory usage, we empirically choose $k \in \{300, 200, 160, 80, 20, 8\}$ depending on the layer size and update type.

---

**Algorithm 2** Genetic Algorithm for Finding Optimal Training Curriculum

---

**Require:** Validation set $\mathcal{D}_{val}$, number of batches $T$, population size $N$, number of generations $K$, mutation probability $p_m$

**Ensure:** Optimal sample order $\pi^{\text{GA}*}$

1: Initialize permutation space $\mathcal{S}_T = \{\pi \mid \pi \text{ is a permutation of } \{1, \ldots, T\}\}$
2: Randomly sample $N$ permutations as initial population: POP $= \{\pi^i\}_{i=1}^N \subset \mathcal{S}_T$
3: **for** $k = 1$ to $K$ **do**
4:     **for all** $\pi^i \in$ POP **do**
5:         Compute $\gamma_T^{\pi^i}$ using FUT with sample order $\pi^i$
6:         Evaluate fitness $r^i = \mathcal{R}(\gamma_T^{\pi^i}, \mathcal{D}_{val})$
7:     **end for**
8:     Retain top $50\%$ individuals with highest fitness to form POP$_{\text{survive}}$
9:     **while** Size of new children $< N/2$ **do**
10:         Randomly select two parents $\pi^a, \pi^b$ from POP$_{\text{survive}}$
11:         Randomly choose crossover points $l, r$ such that $1 \leq l < r \leq T$
12:         Generate child $\pi^c = \text{PMX}(\pi^a, \pi^b, l, r)$
13:         **if** random() $< p_m$ **then**
14:             Randomly select positions $i, j$ and swap $\pi_i^c$ and $\pi_j^c$
15:         **end if**
16:         Add $\pi^c$ to new children
17:     **end while**
18:     Replace discarded individuals in POP with new children
19: **end for**
20: **return** $\pi^{\text{GA}*} = \arg\max_{\pi \in \text{POP}} \mathcal{R}(\gamma_T^{\pi}, \mathcal{D}_{val})$

---

### B.3 GENETIC ALGORITHM FOR TRAINING CURRICULUM DESIGN IN FUT FRAMEWORK

Recall that the objective in equation (9), *i.e.,* $\pi^* := \arg\max_{\pi \in \Pi} \mathcal{R}(\gamma_T^{\pi}, \mathcal{D}_{val})$, is to find the optimal permutation $\pi^*$ that leads to the best validation performance, where $\gamma_T^{\pi}$ represents the final model parameters estimated by FUT framework. However, equation (9) is naturally non-differentiable, hindering its application in finding the optimal curriculum. To address this issue, we design an optimization algorithm based on Genetic Algorithm (GA) (Katoch et al., 2020). GA is a well-established metaheuristic algorithm inspired by Darwinian evolution, which iteratively evolves a population of candidate solutions based on the principle of survival of the fittest. In our context, each candidate represents a specific sample order $\pi$, and the fitness of each individual is evaluated by the model's performance $r^{\pi} = \mathcal{R}(\gamma_T^{\pi}, \mathcal{D}_{val})$. By leveraging crossover, mutation, and selection operators, GA enables us to efficiently explore the exponentially large permutation space without exhaustive enumeration. We describe the detailed design of our GA-based search strategy as follows:

1. **Population Initialization:** Randomly select $N$ sample orders POP$=\{\pi^i\}_{i=1}^N$ from $\mathcal{S}_T$ as the initial populations, where $\mathcal{S}_T = \{\pi \mid \pi \text{ is a permutation of } \{1, \ldots, T\}\}$, with $|\mathcal{S}_T| = T!$.

2. **Fitness Selection:** For each $\pi^i \in$ POP, evaluate the model performance $\mathcal{R}(\gamma_T^{\pi^i}, \mathcal{D}_{val})$ as its fitness, where $\gamma_T^{\pi^i}$ is estimated via the FUT method. Retain the top $50\%$ individuals with the highest fitness scores for reproduction, and discard the rest.

3. **Crossover:** Generate new children by applying the partially matched crossover (PMX) (Kora & Yadlapalli, 2017) to randomly selected parent pairs $\pi^a$ and $\pi^b$ from the surviving population. Specifically, randomly choose two crossover points $l$ and $r$ such that $1 \leq l < r \leq T$, then exchange the subsequences $\pi_{l:r}^a$ and $\pi_{l:r}^b$ between the parents. At last, resolve conflicts using the mapping induced by the swapped segments to produce a valid permutation child $\pi^c = \text{PMX}(\pi^a, \pi^b, l, r) \in \mathcal{S}_T$.

4. **Mutation:** With a predefined mutation probability $p_m$, randomly select two indices $i$ and $j$ in $\pi^c$ and swap their values: $\pi^c \leftarrow \pi_{i \leftrightarrow j}^c$. This operation introduces diversity and prevents premature convergence.

5. **Replacement:** Insert the newly generated children into the population, replacing the discarded individuals. The updated population then forms the basis for the next generation.

By iteratively performing 2-5 steps over a fixed number of generations $K$, or until a convergence criterion is met (e.g., no improvement in validation performance over several generations), the algorithm ultimately returns the best sample order $\pi^{\text{GA}*}$ with the highest validation performance. Therefore, this GA-based optimization reduces the inference time complexity of FUT from $\mathcal{O}(T!)$ to $\mathcal{O}(K \cdot N)$, significantly accelerating the search for the optimal sample order.

## C    EXPERIMENTAL DETAILS

### C.1    GENERAL CAPABILITY

In this section, we introduce more details for the experiments to test the general capability of our FUT framework in Section 5.1.

#### C.1.1    BASE MODEL

We conduct all of our experiments on a language model that follows the LLaMA architecture (Touvron et al., 2023), but with a reduced number of parameters—specifically, a hidden size of 2048 and 10 stacked transformer layers, resulting in approximately 636 million parameters.

We choose this relatively small model to enable repeated training under varying experimental conditions, which is essential for rigorously evaluating the effectiveness of our proposed FUT framework in both training curriculum design and the analysis of memorization and generalization behaviors. In contrast, training large-scale models typically takes tens or even hundreds of days, making such extensive experimentation prohibitively time-consuming and computationally expensive.

#### C.1.2    DATASET

WikiText-103 (Merity et al., 2018a) is a widely used benchmark dataset for evaluating language models, particularly in long-range dependency modeling. It consists of over 100 million tokens extracted from high-quality Wikipedia articles, specifically curated to preserve coherent paragraph and document-level structures. Unlike other common datasets that contain shuffled or sentence-level data, WikiText-103 maintains the original article formatting and ordering, enabling models to better learn contextual and discourse-level information. The vocabulary is relatively large and diverse, making it a challenging and realistic corpus for testing the generalization and memorization capabilities of large-scale language models. In our experiments, we partition the dataset into 80% for training, 10% for validation, and the remaining 10% for testing.

To preprocess the WikiText-103 dataset, we first remove short texts with fewer than five characters to eliminate noise. Then, we apply MinHash-based deduplication (Broder, 1997) to efficiently identify and discard near-duplicate samples. Specifically, each text is tokenized into a set of words, and a MinHash signature is computed using 128 permutations. Texts with identical MinHash digests are considered duplicates, and only one representative is retained. This process effectively reduces redundancy while preserving semantically diverse content.

#### C.1.3    TRAINING AND EVALUATION PROTOCOLS

**Training Protocol.** For the preprocessed WikiText dataset, we split the data into 80%, 10%, and 10% for training, validation, and testing, respectively. The learning rate is selected from the range $[0.0001, 0.005]$ based on validation performance, and we choose the number of batches from $\{8, 16, 32, 64, 128, 256\}$. Since it is not feasible to process very large batch sizes directly due to memory constraints, we apply gradient accumulation over multiple smaller mini-batches to effectively simulate the desired larger batch size. For the Adam optimizer, we fix the hyperparameters $\beta_1$ and $\beta_2$ to 0.9 and 0.95, respectively. Within our FUT framework, to stabilize parameter estimation and mitigate the influence of outliers, we apply parameter clipping. Specifically, the parameters are constrained within a tunable range, with the clipping threshold selected from the interval $[-1.1, -0.3] \cup [0.3, 1.1]$ to ensure numerical stability and prevent extreme values from dominating the update dynamics. The experiments were conducted on a computing platform equipped with NVIDIA A800-SXM GPUs, with a total of 4 GPUs each providing 80GB of memory.

**Evaluation Protocol.** We adopt Perplexity (PPL) (Hu et al., 2024) as the evaluation metric to assess language modeling performance. Given a token sequence $x = (x_1, x_2, \ldots, x_N)$, the perplexity is defined as:

$$\text{PPL}(x) = P(x_1, \ldots, x_N)^{-\frac{1}{N}} = \left( \prod_{t=1}^{N} P(x_t \mid x_{<t}) \right)^{-\frac{1}{N}} = \exp \left( -\frac{1}{N} \sum_{t=1}^{N} \log P(x_t \mid x_{<t}) \right). \tag{16}$$

This is equivalent to the exponential of the average cross-entropy loss. Thus, for a given validation set $\mathcal{D}_{val}$ and final model parameters $\theta_T$, we compute:

$$\text{PPL}(\mathcal{D}_{val}) = \exp \left( \mathcal{L}(\mathcal{D}_{val}; \theta_T) \right), \tag{17}$$

where $\mathcal{L}(\mathcal{D}_{val}; \theta_T)$ denotes the average cross-entropy loss over the validation set.

## C.2 Training Curriculum Design for LLMs

### C.2.1 Baselines

Although curriculum learning largely depends on human heuristics or empirical findings, there are still many works that make efforts to design a rational curriculum in the field of LLMs, primarily based on either the characteristics of the dataset (Campos, 2021), or the quantitative criteria (Zhang et al., 2025a;b) that are perceptible to the model. In this section, we introduce all the baselines used in training curriculum design in detail. For better understanding, we define $\rho_{B_i}$ as the difficulty score for batch $B_i$.

• **Random Order (RO).** RO is a naive baseline, which randomly assigns the difficulty score $\rho_{B_i}$ to each batch $B_i$ in the range of $[0, 1]$.

• **Sample Length (SL) (Campos, 2021).** SL is a purely statistical method based on the intuition that longer sentences are inherently more difficult to model. This is because they require more effective tracking of dependencies, making the learning process more challenging. Therefore, the difficulty score of each batch $B_i$ is defined as the total number of tokens in the batch, computed as $\rho_{B_i} = \sum_{x \in B_i} |x|$, where $|x|$ denotes the length of sample $x$.

• **Perplexity (PPL) (Zhang et al., 2025a).** PPL metric closely aligns with the self-supervised learning objective of LLMs and effectively measures model-data fit, making it appropriate for data organization. Recent studies (Zhang et al., 2025a) empirically show that training on high-PPL data followed by low-PPL data can significantly reduce loss and boost performance. Following this finding, we introduce a reference model $M_{ref}$ with parameter $\theta_R$ to compute PPL for each batch as the difficulty score, *i.e.,* $\rho_{B_i} = -\mathcal{R}(\theta_R, B_i)$.

• **Perplexity Difference (PD) (Zhang et al., 2025b).** Building on the idea in (Zhang et al., 2025b), PD between strong and weak models can serve as an indicator of how difficult a batch is for the model. Specifically, a low PD implies that both models perform similarly in terms of learning efficiency, while a high PD suggests that the batch presents greater difficulty for the weaker model. Consider two reference models, $M_{str}$ and $M_{weak}$, with parameters $\theta_S$ and $\theta_W$, respectively, both trained on the same dataset. In practice, we train two models: $M_{str}$ with 636 million parameters and $M_{weak}$ with 167 million parameters, using their perplexity differences to guide batch rescheduling. For each batch $B_i$, we define PD as the difficulty score, given by $\rho_{B_i} = (\mathcal{R}(\theta_W, B_i) - \mathcal{R}(\theta_S, B_i))/\mathcal{R}(\theta_W, B_i)$.

### C.2.2 Genetic Algorithm Configuration

To effectively search the optimal sample order within the exponentially large permutation space, we employ a Genetic Algorithm (GA) tailored to our FUT framework. The key design choices focus on maintaining a balance between exploration and exploitation: a moderately sized population ensures sufficient diversity, while elitist selection preserves high-quality solutions across generations. The complete set of hyperparameters and their configurations are summarized in Table 3.

Table 3: Genetic Algorithm hyperparameters used in our framework

| Hyperparameter | Notation | Description | Scope |
|---|---|---|---|
| Population size | $N$ | Number of candidates per generation | $[16, 12, 8, 4, 2]$ |
| Max generations | $K$ | Total evolution rounds | $[16, 12, 8, 4, 2, 1]$ |
| Number of batches | $T$ | Total number of Batches | $[256, 128, 64, 32, 16, 8]$ |
| Crossover points | $l, r$ | Random crossover segment indices | $1 \leq l < r \leq T$ |
| Mutation probability | $p_m$ | Swap probability per child | $0.1$ |
| Selection rate | $-$ | Top individuals retained | $50\%$ |

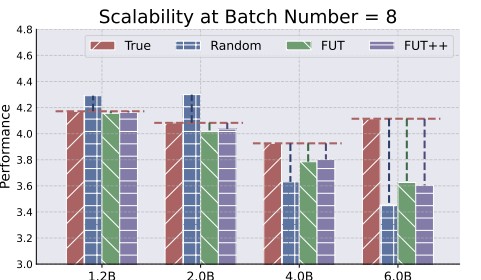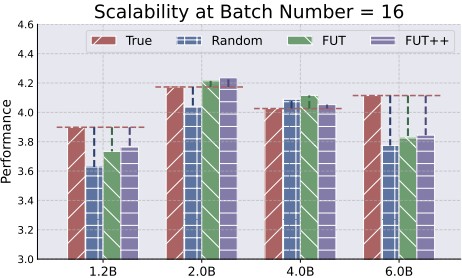

Figure 6: **Scalable estimation performance across model sizes.** We evaluate the estimation accuracy of FUT and FUT++ across model scales $\{1.2B, 2.0B, 4.0B, 6.0B\}$ under training batch numbers $T = 8$ (left) and $T = 16$ (right). FUT and FUT++ consistently outperform the Random baseline, with FUT++ showing improved accuracy for larger models.

# D    ADDITIONAL EXPERIMENTAL RESULTS

## D.1    SCALABILITY OF FUT FRAMEWORK

**Experimental Setup.** In this section, we conduct additional experiments to evaluate whether our proposed FUT framework remains effective in estimating model performance as the base model size increases. Specifically, we scale the original $0.6B$ model to $\{1.2B, 2.0B, 4.0B, 6.0B\}$. In these experiments, the number of training batches is set to $T = 8$ and $T = 16$. We adopt perplexity as the evaluation metric and measure the performance gap between the true values and the estimates produced by our FUT framework.

**Results.** The results are illustrated in Figure 6. Across both batch settings ($T = 8$ and $T = 16$), our proposed FUT and FUT++ methods consistently outperform the Random baseline in estimating model performance, achieving smaller performance gaps to the ground truth. This trend holds true as we scale the base model size from $1.2B$ to $6.0B$, validating the scalability of our framework. Importantly, we observe that FUT++—which incorporates second-order information—yields even more accurate estimates compared to the original FUT, particularly for larger models. This suggests that higher-order approximations are more effective at capturing complex parameter updates in large-scale language models. The Random baseline, by contrast, lacks theoretical grounding and exhibits less consistent performance as model size grows.

## D.2    IMPACT OF RANDOM PROJECTION TECHNIQUE

**Experimental Setup**. Random projection is used in our FUT framework to squeeze the storage complexity, which makes the performance estimation of large model with massive parameters become possible. In this section, we aim to explore how random projection can affect the final estimation accuracy. Since the entire storage of pairs of update terms is unaffordable for a single machine, we conduct this ablation study in a relatively small model with 200M parameters. All other experimental setup is consistent with the main experiment.

**Results.** The results are presented in Tale 4, where we can see that the use of random projection may have negative impact on estimation accuracy, because the variant without using random projection have a more accurate estimation to the true perplexity performance. Nevertheless, we can

Table 4: Impact of random projection. We experiment on all different number of batches from 8 to 256 in model with 200M parameters. The results show that the use of random projection can inevitably hurt the estimation accuracy, but it can largely save the storage memory.

| Batch Num | Accuracy | | | Memory | |
|---|---|---|---|---|---|
| | True | With RP | W/O RP | With RP | W/O RP |
| 8 | 1.4885 | 1.5158 | **1.5016** | $\sim 0.3$G | $\sim 12$G |
| 16 | 1.4569 | 1.4793 | **1.4789** | $\sim 1.6$G | $\sim 52$G |
| 32 | 1.4512 | 1.4839 | **1.4783** | $\sim 6.3$G | $\sim 200$G |
| 64 | 1.4485 | **1.4468** | 1.4647 | $\sim 26$G | $\sim 820$G |
| 128 | 1.4359 | 1.4269 | **1.4394** | $\sim 102$G | $\sim 3.2$T |
| 256 | 1.4268 | **1.4258** | 1.4358 | $\sim 410$G | $\sim 13.1$T |

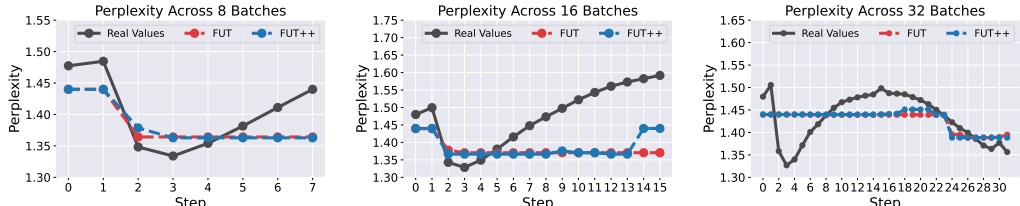

Figure 7: **Perplexity estimation at intermediate training steps.** We visualize the validation perplexity estimated by FUT and FUT++ compared to the real validation perplexity after each batch, for training schedules with $T \in \{8, 16, 32\}$ total batches. FUT and FUT++ both closely follow the true performance trends, with FUT++ consistently providing more accurate estimates—especially when $T$ is larger. These results demonstrate the effectiveness of our methods in tracking training progress in a fine-grained manner.

observe that the use of random projection can largely save the storage memory, especially when the number of batches increases. Therefore in practice, random projection can still have great function in implementing our framework.

### D.3 BATCH-WISE ANALYSIS OF PERFORMANCE ESTIMATION ACCURACY

**Experimental Setup.** In this section, we conduct a fine-grained evaluation of our FUT framework by comparing estimated and true model performance at intermediate stages of training. Specifically, we consider batch numbers $T \in \{8, 16, 32\}$ and evaluate performance after each training batch. For each time step $1 \leq t \leq T$, we replace the final-step performance comparison $\mathcal{R}(\gamma_T^\pi, \mathcal{D}_{val})$ and $\mathcal{R}(\theta_T^\pi, \mathcal{D}_{val})$ with the intermediate-step comparison $\mathcal{R}(\gamma_t^\pi, \mathcal{D}_{val})$ and $\mathcal{R}(\theta_t^\pi, \mathcal{D}_{val})$. We use perplexity on the validation set $\mathcal{D}_{val}$ as the evaluation metric to assess how well the FUT-estimated parameters align with those obtained from actual training at each step.

**Results.** As shown in Figure 7, both FUT and FUT++ generate accurate perplexity estimates across different training stages. While FUT performs well in general, FUT++ shows higher fidelity—especially as the number of batches increases. This is most evident in the $T = 32$ case, where FUT++ remains close to the true perplexity throughout, whereas FUT slightly deviates in later stages. These findings affirm the utility of incorporating higher-order dynamics in FUT++, and highlight the robustness of our framework in real-time model monitoring, dynamic training adaptation, and early stopping decisions. In addition, we observe that in certain training stages, particularly under small batch sizes or early steps, the estimated perplexity remains unchanged over multiple steps, forming plateau-like segments. This phenomenon arises from the Taylor-based approximation mechanism in our framework. Specifically, when the update gradients are small (*e.g.,* due to flat regions in the loss landscape), the computed updates become negligible. Consequently, FUT and FUT++ produce nearly identical estimates across consecutive steps.

# E    FURTHER DISCUSSIONS OF RELATED WORK

## E.1    CURRICULUM LEARNING FOR LLMS

Curriculum learning is a training paradigm that organizes training data in an easy-to-hard manner to facilitate more effective learning (Bengio et al., 2009; Graves et al., 2017; Hacohen & Weinshall, 2019; Xu et al., 2020). In deep learning tasks, sample difficulty is typically defined using either surface-level heuristics or model-based metrics (Matiisen et al., 2019; Hacohen & Weinshall, 2019; Gui et al., 2017; Ghebrechristos & Alaghband, 2020; Weinshall et al., 2018). For instance, in sequence modeling, easier examples are often shorter or contain more frequent tokens (Zhang et al., 2018a). In the generative modeling domain, difficulty can be measured by how well a sample aligns with human cognitive expectations or its deviation from the data distribution center (Tudor Ionescu et al., 2016; Zhao et al., 2019). In the context of LLMs, several empirical studies have explored strategies to score training samples (Naïr et al., 2024; Liang et al., 2024; Wang et al., 2024; Matiisen et al., 2019; Campos, 2021; Zhang et al., 2025a;b). Specifically, (Campos, 2021) reorganizes samples based on their sequence length to progressively improve the model's ability to capture long-range dependencies. Furthermore, some researchers (Zhang et al., 2025a;b) propose curriculum schemes guided by model-based metrics such as perplexity and perplexity difference, motivated by their empirical observations. **In contrast to conventional curriculum learning approaches that depend on human-designed heuristics for determining sample order, our proposed FUT framework offers an efficient and reliable means of estimating final performance across arbitrary curricula.** This allows practitioners to make well-informed decisions among diverse curriculum strategies without incurring the cost of repeated retraining.

## E.2    ZEROTH-ORDER OPTIMIZATION

Zeroth-order (ZO) optimization refers to a class of derivative-free methods that estimate gradients using only function evaluations, making them suitable for black-box or simulation-based scenarios where gradients are inaccessible or costly (Flaxman et al., 2004; Ghadimi & Lan, 2013; Nesterov & Spokoiny, 2017; Duchi et al., 2015; Wang et al., 2018; Chen et al., 2017; Liu et al., 2020). Classical approaches include finite-difference methods (Flaxman et al., 2004), random gradient estimators (Nesterov & Spokoiny, 2017; Duchi et al., 2015), and ZO-SGD (Ghadimi & Lan, 2013). Recently, ZO has been applied to LLM fine-tuning to reduce the memory burden of back-propagation. Notably, MeZO (Malladi et al., 2023) introduced a forward-only ZO-SGD variant, while Zhang et al. (Zhang et al., 2024) benchmarked and extended ZO techniques—such as ZO-Adam (Chen et al., 2019b) and block-wise estimation—for scalable LLM fine-tuning. **However, applying ZO to pre-training remains impractical due to the extreme dimensionality of LLMs, high variance of estimators, and computational overhead from repeated forward passes** (Zhang et al., 2023; Golovin et al., 2019; Wang et al., 2018). Moreover, most of ZO methods rely on dynamic random perturbations, limiting result reproducibility and reuse. **In contrast, our FUT framework is a performance estimation tool—not an optimizer—that precomputes all necessary update terms using Taylor expansions.** This enables efficient, deterministic evaluation of arbitrary curricula without retraining, making FUT quite suitable for analyzing training dynamics and guiding curriculum design.

# F    BROADER IMPACTS

With the rapid advancement of LLMs, not only have their language understanding and reasoning abilities improved, but their parameter sizes have also grown significantly. As a result, training LLMs has become increasingly time-consuming and computationally expensive. In this paper, we propose a retrain-free framework called FUT, which accurately estimates model performance using Taylor expansion. This has several important practical implications.

First, FUT enables researchers to study the effect of training sample order on LLM performance without repeated retraining, including downstream applications such as memorization and generalization analysis. The performance estimates associated with different sample orders provide valuable insights into both internal learning dynamics and external behavior.

Second, the FUT framework can serve as a tool for efficient performance evaluation and training analysis in large-scale model development pipelines. For example, developers can leverage FUT to screen and prioritize data curricula, identify critical samples, or detect unstable training configurations before committing to full-scale training. As LLMs continue to scale, such cost-effective analysis tools will be increasingly essential to accelerate research while reducing resource consumption.

## G  LIMITATIONS

While our proposed retraining-free framework (FUT) provides a computationally efficient and theoretically grounded method for estimating the effects of sample order in LLMs, several limitations should be acknowledged.

1. The accuracy of our estimates relies on the validity of Taylor expansions, particularly when higher-order nonlinearities dominate the optimization dynamics—scenarios where our first- and second-order approximations may fall short.

2. Although the use of random projection significantly reduces memory overhead, it may introduce approximation noise, especially for models with extremely large parameter spaces.

3. We evaluate the effectiveness of our FUT framework solely based on perplexity performance. This is because downstream natural language understanding and reasoning tasks typically require large-scale models, which are infeasible to retrain repeatedly under varying conditions. Nevertheless, further validation is needed to assess the generalizability of our framework in these more complex tasks.

