# OpenReview forum: "Estimating the Effects of Sample Training Orders for Large Language Models without Retraining"
_ICLR.cc/2026/Conference — ICLR 2026 Conference Withdrawn Submission_

### Official Review · Reviewer_tW7G · 2025-10-22

**Soundness:** 2
**Presentation:** 3
**Contribution:** 2
**Rating:** 4
**Confidence:** 4

**Summary:**

The paper introduces FUT, a retraining-free framework to estimate the performance of Large Language Models (LLMs) under arbitrary training sample orders. The core method approximates the Adam optimizer's trajectory using first- and second-order Taylor expansions around a pre-computed reference trajectory. To manage the significant storage requirements of the necessary gradient terms, the framework employs random projection. The authors demonstrate the utility of FUT on two applications: finding optimal training curricula and analyzing the effects of sample position on memorization and generalization, with all evaluations conducted based on perplexity.

**Strengths:**

*   The paper addresses a critical and practical problem in LLM training. Evaluating the impact of data ordering without incurring the prohibitive cost of full retraining is a highly valuable research direction.
*   The proposed FUT framework is technically well-grounded. Deriving estimations from a mathematical approximation of the optimization dynamics via Taylor expansions provides a clear and interpretable foundation, moving beyond purely heuristic methods.
*   The framework's methodology, particularly its division into three distinct stages (Reference Training, Update Term Storing, and Estimation), is presented clearly and is easy to follow.

**Weaknesses:**

*   The empirical evaluation is solely based on perplexity (PPL). It remains unsubstantiated whether a PPL-optimal training order, as identified by FUT, translates to improved performance on established downstream benchmarks (e.g., MMLU, GSM8K).
*   The experimental comparison is weak. FUT is primarily compared against full retraining and a simple random baseline. The paper omits a crucial comparison with more relevant and advanced methods for retraining-free analysis, such as modern, Hessian-free influence functions (e.g., TracIn) or proxy model-based approaches.
*   The justification for the substantial precomputation cost, $O(T^2 \cdot C)$, is tenuous. The claimed benefit rests on finding a PPL-optimal curriculum, whose value for downstream tasks is not demonstrated, making the high upfront investment difficult to justify from a practical standpoint.

**Questions:**

see weakness

---

### Official Review · Reviewer_wMp4 · 2025-10-28

**Soundness:** 3
**Presentation:** 2
**Contribution:** 2
**Rating:** 4
**Confidence:** 3

**Summary:**

This paper introduces a retraining-free framework. It approximates Adam optimizer updates with first and second-order Taylor expansions and the author utilize it to find the best curriculum design for LLM training. The author theoretically demonstrates that it has better order compared against greedy search and it achieves better performance over baselines.

**Strengths:**

1. The intuition behind this paper is easy to understand.
2. The illustration is clear and helpful.

**Weaknesses:**

1. The improvement is limited. From Table 2, we can see that comparing against baselines, FUT and FUT++ improves a little. Considering the timing cost of FUT, I am not sure if we should utilize this algorithm in practice.
2. The method is tested in limited settings. I have doubt if the method is robust across different model sizes, model architectures and dataset distributions. The effectiveness of the algorithm should be tested on various scenarios.
3. In Table 2, the author is encouraged to list time/memory cost for each baselines and algorithms. I have doubt if the comparison is fair if we only focus on the final performance.

**Questions:**

Please check the weakness part.

---

### Official Review · Reviewer_8Whi · 2025-10-29

**Soundness:** 1
**Presentation:** 2
**Contribution:** 2
**Rating:** 2
**Confidence:** 3

**Summary:**

This paper addresses the computationally prohibitive cost of studying the effect of training sample order on Large Language Models (LLMs), which traditionally requires a full retraining for each new permutation. The authors propose a retraining-free framework, FUT, to estimate the model parameters for any new sample order. The method works by first training a model on a single reference order to get a trajectory of checkpoints. It then precomputes and stores all optimizer update terms and their first and second-order gradients for all pairs of reference checkpoints and training batches. To estimate the parameters for a new order, it recursively applies the optimizer update, approximating the update term using a first-order (FUT) or second-order (FUT++) Taylor expansion around the reference checkpoint. To manage the massive storage cost, the matrices of update terms are compressed using Random Projection. The framework is applied to two downstream tasks: training curriculum design (using a Genetic Algorithm that uses FUT for a fitness function) and analyzing LLM memorization and generalization effects.

**Strengths:**

- **Problem Significance:** The paper tackles a well-defined, important, and challenging problem. Developing a retraining-free method to analyze sample order would be a major contribution to understanding LLM training dynamics, curriculum learning, and interpretability.
- **Novel Problem Formulation:** The paper provides a clear problem formulation in Section 2, formally defining the goal of estimating the parameter trajectory for a new sample order based on a reference trajectory.

**Weaknesses:**

- **Scalability Issue:** The framework's core premise relies on a precomputation and storage step that is $\mathcal{O}(T^2)$ in $T$, the number of training batches. This complexity is fundamentally non-scalable and practically infeasible for any realistic LLM training scenario, where $T$ is often in the hundreds of thousands or millions. The paper's own data in Appendix D.2 (Table 4) confirms this: storing the *uncompressed* update terms for a small 200M model with only $T=256$ batches already requires **13.1 Terabytes** of storage. Even with lossy random projection, the storage is 410GB.
- **Unstable Approximation and Compounding Error:** The method's core approximation (Taylor expansion around $\theta_t$ to estimate the state at $\gamma_t$) is unstable. As the new and reference trajectories diverge (which they must, given different data orders), the approximation error grows at each step. Figure 7 provides clear evidence of this failure, showing the FUT/FUT++ estimates "flat-lining" and completely failing to track the real model's dynamics. The claim to analyze "learning dynamics" is therefore unsupported.
- **Approximation of High-Order Gradients:** The FUT++ method, claimed to be superior, requires second and third-order derivatives of the loss ($\nabla^2 \mathcal{L}$, $\nabla^3 \mathcal{L}$). The paper states these are approximated using finite differences (e.g., $\nabla^2 \mathcal{L} \approx (\nabla \mathcal{L}(\theta_t) - \nabla \mathcal{L}(\theta_{t-1})) / (\theta_t - \theta_{t-1})$). This is a very rough approximation, which is then *stacked* (using the approximated $\nabla^2 \mathcal{L}$ to approximate $\nabla^3 \mathcal{L}$). This introduces multiple, unquantified layers of approximation error on top of the already-unstable Taylor expansion.

**Questions:**

1. **Validity of Taylor Approximation and Error Propagation.** In Figure 7, the estimated perplexity from FUT/FUT++ diverges significantly from the "Real Values," often flat-lining while the true perplexity fluctuates. This suggests a cascading approximation error builds up as the estimated trajectory $\gamma_t$ diverges from the reference trajectory $\theta_t$. How can the final performance estimates (e.g., in Table 2) be considered reliable if the intermediate parameter trajectory is demonstrably incorrect?
2. **Accuracy of Computational Cost Analysis.** The complexity analysis of $\mathcal{O}(T^2 \cdot C)$ seems to underestimate the true cost.
    1. First, what is the cost of the Estimation Stage (Stage 3 in Algorithm 1 3), and is it truly negligible compared to the $\mathcal{O}(T^2 \cdot C)$ precomputation?
    2. Second, the analysis assumes computing higher-order gradients (like $\nabla^2 \mathcal{L}$ needed for FUT++) has the same cost $C$ as the first-order gradient. However, computing Hessians is far more expensive. The paper mentions approximating this in Appendix B.1, but the cost of this approximation is not clearly included in the $\mathcal{O}(T^2 \cdot C)$ estimate. Could you clarify the practical cost of computing these higher-order terms and update the complexity analysis?
3. **Justification for Johnson-Lindenstrauss (JL) Theorem.** The paper states it uses the random projection to reduce storage. However, it is unclear what is the usage of JL theorem, i.e. preservation of the pairwise distances, in the proposal.
4. **Clarity of Error Metric (Table 1).** Table 1 reports performance using `AbsDiff` (Absolute Difference). This metric can be misleading, as an `AbsDiff` of 0.02 on a true perplexity of 1.4 (as in T=256 ) is a small relative error, while the same `AbsDiff` on a perplexity of 0.05 would be much larger. Could the authors also report a relative error metric to better contextualize the significance of the estimation accuracy?
5. **Quantitative Comparison for Memorization (Figure 3).** Figure 3 presents the memorization analysis as three heatmaps (FUT, FUT++, and True). While visually similar, this comparison is subjective. Could you provide a quantitative comparison, such as the Mean Squared Error (MSE) or L1 norm between the estimated heatmaps (a, b) and the true heatmap (c), to rigorously evaluate the accuracy of the memorization analysis?
6. **Effect of batch-size**. It would be better to discuss the effect of batch size (or the number of batches T) on the performance, estimation error, and efficiency.
7. **Applicability to Multi-Epoch Training**. The application for training curriculum design (Section 4.1) focuses on finding an optimal order for a single epoch. This is not a common setting for curriculum learning, which typically involves data scheduling over multiple epochs. How could the FUT framework be extended to estimate performance for multi-epoch training, where the data used for different epochs might be different?
8. **Definition of Memorization.** The paper defines the memorization effect $M_{i,j}$ as the model's performance on batch $B_i$ when $B_i$ is fixed at position $j$ in the training sequence. This seems to measure recency bias (i.e., how well the model remembers $B_i$ based on *when* it saw it), which is more akin to catastrophic forgetting in continual learning. This differs from common definitions of memorization, which often involve measuring the influence of a sample's *presence* versus its *absence*. Could the authors justify this definition and clarify its relationship to other established definitions of memorization?

---

### Official Review · Reviewer_49LT · 2025-10-30

**Soundness:** 2
**Presentation:** 3
**Contribution:** 2
**Rating:** 4
**Confidence:** 3

**Summary:**

This paper introduces FUT, a framework designed to estimate the effects of training data order on LLMs without the prohibitive cost of full retraining. The core idea is to leverage a single reference training run to approximate the parameter trajectories of any new data permutation. This is achieved by recursively applying a first- or second-order Taylor expansion to the Adam optimizer's update rule, using pre-computed gradient terms. To manage the otherwise infeasible computational and storage costs, the framework relies on a cascade of approximations, including finite differences for higher-order gradients and Random Projection for compressing the stored terms. The authors demonstrate the framework's utility on two applications: designing training curricula and analyzing model memorization and generalization.

**Strengths:**

The paper's primary strength is its novel attempt to tackle the intractable problem of data ordering effects in LLM training. The proposed FUT framework is theoretically principled, grounded in Taylor expansions, and offers a new approach to estimate the effect of data curriculum. The empirical results, while limited in scale, demonstrate that the method can outperform a naive baseline in its estimation task and can be used to design curricula that are superior to common heuristics.

**Weaknesses:**

1. **Questionable Practicality and Scalability:** The paper's claim of "computational efficiency" is based on an amortized analysis that masks an extremely impractical workflow for large-scale LLM pre-training. The method requires a full, costly reference run and a subsequent pre-computation stage with $\mathcal{O}(T^2 \cdot C)$ complexity. The experiments are conducted on a maximum of $T=256$ batches, a scale that is trivial compared to the millions of steps in a typical pre-training run. At that scale, the pre-computation and storage (even with Random Projection) would be completely infeasible. This massive gap between the demonstrated scale and the target application domain makes the primary use case of curriculum design for pre-training appear unrealistic.
2. **Fragility of Compounding Approximations:** The framework's accuracy rests on a chain of approximations, each introducing a source of error. The core recursive estimation, based on a local Taylor approximation, is fundamentally vulnerable to compounding error. Any small deviation between the estimated trajectory ($\gamma_t$) and the reference trajectory ($\theta_t$) will degrade the approximation's quality, and this error will propagate and likely grow exponentially at each subsequent step. Over a realistic training horizon, this accumulated error would likely to overwhelm the signal, rendering the final estimate meaningless. The use of finite difference to approximate the Hessian adds another layer of potential numerical instability, which is then further distorted by the lossy Random Projection compression. The paper does not adequately analyze the interaction and compounding of these error sources.

**Questions:**

The paper presents an interesting theoretical idea, but significant limitations in practical application undermine its contribution. To strengthen the work, I suggest the following directions:

1. **Reframe the contribution and demonstrate more compelling use cases.** Given the severe scalability issues, the claim of providing a practical tool for LLM curriculum design is not well-supported. The authors should consider repositioning the paper's primary contribution as a "scientific instrument" for small-scale research and development rather than a practical large-scale tool. More importantly, the paper would benefit from applying the framework to answer concrete, open research questions uniquely suited to its capabilities. Providing even small-scale empirical evidence for questions like those below would offer a far more compelling demonstration of value than the current applications:
- Does placing difficult, domain-specific data (e.g., legal text) early versus late in training have a larger impact on generalization?
- Is there a specific sequence of data domains (e.g., web text → books → code) that consistently yields better models?
- How does the placement of a single "poisoned" or low-quality data batch affect the final model's safety and alignment?

2. **Address robustness and error propagation rigorously.** The paper lacks critical analysis of the framework's limitations. An ablation study examining sensitivity to the reference trajectory is notably absent. Additionally, the authors need to provide either theoretical or empirical analysis of error accumulation bounds (even for a simplified case) to demonstrate that the method does not simply devolve into noise over a meaningful number of steps.

---

### Note · Authors · 2025-11-23

I have read and agree with the venue's withdrawal policy on behalf of myself and my co-authors.